# Mitochondria Drive Immune Responses in Critical Disease

**DOI:** 10.3390/cells11244113

**Published:** 2022-12-18

**Authors:** Shilpa Tiwari-Heckler, Simon C. Robson, Maria Serena Longhi

**Affiliations:** 1Department of Gastroenterology, University Hospital Heidelberg Medical Clinic, 69120 Heidelberg, Germany; 2Center for Inflammation Research, Department of Anesthesia, Critical Care and Pain Medicine, Beth Israel Deaconess Medical Center, Harvard Medical School, Boston, MA 02215, USA; 3Division of Gastroenterology, Department of Medicine, Beth Israel Deaconess Medical Center, Harvard Medical School, Boston, MA 02215, USA

**Keywords:** mitochondria, innate immunity, adaptive immunity, immunometabolism, mitochondrial danger-associated-molecular-pattern, inflammation, immunosuppression, trauma, critical illness

## Abstract

Mitochondria engage in multiple cellular and extracellular signaling pathways ranging from metabolic control, antiviral and antibacterial host defense to the modulation of inflammatory responses following cellular damage and stress. The remarkable contributions of these organelles to innate and adaptive immunity, shape cell phenotype and modulate their functions during infection, after trauma and in the setting of inflammatory disease. We review the latest knowledge of mitochondrial biology and then discuss how these organelles may impact immune cells to drive aberrant immune responses in critical disease.

## 1. Introduction

Recent advances have highlighted the pleiotropic functions of mitochondria in modulating cell metabolism and key signaling pathways during infection and inflammation. Mitochondria are the powerhouse of the cell, providing energy through the generation of adenosine triphosphate (ATP) from oxidative phosphorylation (OXPHOS). Several studies, moreover, have suggested multiple other roles of mitochondria, impacting metabolism and immune responses while altering cell phenotype and modulating function in the context of host defense and inflammation [1,2,3]. 

In this review we will discuss the pathophysiology of mitochondria and highlight innate and adaptive immune responses induced by or related to mitochondrial functionality. Further, we will focus on the significant involvement of mitochondria in driving critical illness. 

## 2. Mitochondrial Biology

It has been suggested that the mitochondrion, originally derived from bacteria, was phagocytosed by proto-eukaryotic cells forming a symbiosis. Accordingly, the structure of mitochondria has overlapping features with those of bacteria. Both comprise an inner and an outer membrane separated by an intermembrane space. In mitochondria, porins are located in the outer layer, which allows the diffusion of proteins <5000 Da [1,4]. Acetyl-CoA derived from pyruvate metabolism or fatty-acid oxidation enters the mitochondria to initiate the tricarboxylic acid (TCA) cycle, also known as Krebs cycle. Important steps of the Krebs cycle occur in the mitochondria matrix to produce nicotinamide adenine dinucleotide (NADH), a molecule that fuels the electron transport chain to ultimately generate ATP through OXPHOS [1,4]. 

The electron transport chain is in the inner mitochondria membrane and consists of four respiratory chain proteins (Complex I-IV), which regulate the electron transport to complex V. Complex V is an ATP synthase, which phosphorylates adenosine diphosphate (ADP) to ATP [1,4]. This OXPHOS mechanism in the mitochondria generates 32 ATP per glucose molecule under normoxic conditions. ATP can also be produced by glycolysis, a metabolic process initiated during an anaerobic state, like hypoxia, which converts glucose to pyruvate and then generates lactate. This step is much faster, but the yield is significantly lower because one glucose molecule ultimately only generates two new ATP molecules [1,3,5]. Thus, mitochondrial ATP production is dependent on sufficient oxygen supply and mitochondrial metabolism can be altered under hypoxic conditions [1,4]. The interactions between anaerobic/aerobic state, cell metabolism and immune cell function represent a new emerging field of investigation termed immunometabolism [6,7]. 

Mitochondria have their own genetic code, mtDNA, located within the matrix and encoding for 13 proteins of the respiratory chain, 22 transfer RNA (tRNA) and 2 ribosomal RNA (rRNA). Furthermore, the genetic information of almost 1500 mitochondrial proteins required for the replication and expression of mtDNA or for OXPHOS machinery are stored in the cell nucleus [8,9]. 

The crosstalk between mitochondria and the cell nucleus is crucial for adequate mitochondria and cell function [9,10]. To maintain this communication, mitochondria build a tubular network through interactions with the endoplasmic reticulum (ER) and undergo regular fusion to combine with other mitochondria; or fission to separate and form new mitochondria [1,9,11]. The formation of mitochondria-ER interaction is dependent on actin filaments. Moreover, mitochondrial kinetics as well as apoptosis rely on actin [11,12]. Studies on mitochondrial trafficking primarily conducted in neurons have provided a quantitative measurement of the mitochondrial transport rate that ranges from 0.4 μm/min to 1 μm/min [11,13]. Additional work has suggested that mitochondrial recruitment depends on energy demand and might be even associated with membrane potential. Active mitochondria move to regions characterized by high energy requirements, while impaired mitochondria gather in cell soma for destruction or repair [11,13]. 

Mitochondrial fission protein 1 (FIS1) and dynamic-related protein (DRP1) control fission processes, whereas Mitofusion proteins (MFN1/2) are regulators of mitochondrial fusion [9,11]. MFN1/2 are expressed on the outer mitochondrial membrane and mediate ER-mitochondria interactions [11]. The balance between fusion and fission processes has major relevance not only for determining the cell number, but also for antiviral host response and for the generation of reactive oxygen species (ROS) that will be discussed in the following section. 

In addition to the generation of ATP, the mitochondrial oxidative metabolism contributes to the production of significant amount of cellular ROS, an important signaling molecule in various redox-sensitive signaling pathways [14]. 

## 3. Interplay between Mitochondria and Innate Immune Cells

Mounting evidence suggests that mitochondria modulate innate immunity. Immune responses can be mediated by mitochondrial metabolites or mitochondrial-derived components, eliciting key signaling pathways during infection and sterile inflammation, as detailed below. 

The activation of innate immunity is one of the first-line host defenses against bacterial and viral infections. Pathogens are recognized by pattern recognition receptors (PRRs) including, amongst others, Toll-like receptors (TLRs), NOD-like receptors (NLRs) and retinoic acid-inducible gene-I (RIG-I)-like receptors [14,15]. These receptors sense molecular fractions presented as pathogen associated molecular pattern (PAMPs) and rapidly activate downstream signaling pathways that modulate cell functional phenotype and enable cells to defend against pathogens [15]. PRRs can also be activated by non-bacterial alarmins, called danger associated molecular pattern (DAMPs), which can be released during cellular stress or apoptosis. During this insult the mitochondrial membrane integrity is altered, and mitochondrial pores are formed, leading to leakage of mitochondrial components into the cytosol or the extracellular space [2,16] ATP release into the extracellular milieu can occur during pathological conditions, such as inflammation, hypoxia or apoptosis [17]. Extracellular ATP is rapidly converted into anti-inflammatory adenosine by a two-step enzymatic reaction regulated by ectoenzymes. Ecto-nucleoside triphosphate diphosphohydrolases (E-NTPDases), including E-NTPDase 1 (CD39), convert ATP/ADP into AMP, and ecto-5′-nucleotidase (CD73) produces extracellular adenosine from AMP [18,19]. Mounting evidence proposes an important link between hypoxia and purinergic signaling. In this regard, several studies have indicated the importance of extracellular adenosine that increases the expression of ectoenzymes during hypoxic conditions [19,20]. Eltzschig et al. have demonstrated that CD39 is upregulated by the transcription factor Sp1 during hypoxia, while Synnestvedt et al. found that CD73 is a transcriptional target of hypoxia-inducible factor 1-alpha (HIF-1α) [21,22]. For a more comprehensive review on the subject we refer to the review article by Bowser et al. [19].

As mentioned before, due to the proposed origin, bacteria and mitochondria share similarities in architecture and biochemistry [1]. Mitochondrial DNA consists of repeated CpG codes, which is also present in bacteria, and which is known to be a potent activator of TLR9. Binding of extracellular mtDNA to TLR9 on neutrophils initiates a rapid proinflammatory response, which is characterized by high expression of Interleukin (IL)-6 and Tumor necrosis factor (TNF) induced upon activation of NF-κB signaling [23]. The cytosolic mtDNA is sensed by the enzyme cyclic guanosine monophosphate–adenosine monophosphate (cGAMP) synthase (cGAS), which, in turn, is responsible for the generation of the second messenger cGAMP. In turn, cGAMP activates the ER resident adaptor Stimulator of Interferon Genes (STING). 

This above interaction leads to the recruitment and activation of TANK-binding kinase 1 (TBK1). TBK1 phosphorylates Interferon regulatory factor 3 (IRF3), which mediates the expression of Interferon (IFN)-β and promotes type I IFN responses following viral infection [14,15,24]. Recent work by Lood et al. has indicated that mtDNA from neutrophils activates the cGAS-STING pathway to modulate type I IFN response in neutrophils in lupus-like disease [14,25]. Thus, several studies have demonstrated that the activation of the cGAS-STING pathway by mtDNA plays important roles in infectious diseases and autoimmunity [14].

Another pathway engaged by mtDNA is that of “Inflammasome-related signaling”. Recent studies have shown that mtDNA activates and interacts with the NLR family pyrin domain containing 3 (NLRP3) inflammasome, which results in the activation of caspase 1, IL-1β and IL-18 cytokine secretion with pyroptic cell death signaling [26]. Collectively, a wealth of studies has shown the importance of mtDNA as a DAMP and its crucial role in regulating infections and inflammation. However, it remains unclear what determining factors drive certain PRRs and pathway activation in response to mtDNA. It appears to elicit different immune responses in a tissue- and cell- specific manner [15]. 

ATP is the central product of mitochondria and also a known activator of NLRP3 inflammasome following engagement of the released nucleotide with P2X purinoceptor 7 (P2X7) and activated purinergic signaling responses. A study by Iyer al. showed that ATP derived from intact extracellular mitochondria per se can further enhance the inflammasome-mediated IL-1β and IL-18 production in innate immune cells [9,27]. 

In addition to mitochondrial DNA and ATP, mitochondrial derived ROS may also contribute to the NLRP3 inflammasome signaling [9,28]. Some studies have indicated that reduction of mROS by treatment with various antioxidants is followed by a substantial inhibition of inflammasome pathways [14,28]. Correspondingly, an increase in mROS production through OXPHOS inhibition in macrophages causes NLRP3-dependent activation of caspase-1 while enhancing IL-1β and IL-18 secretion [29]. Moreover, exposure of mitochondria targeted antioxidants in OXPHOS-deficient macrophages results in decreased NLRP3 inflammasome activation [14,30]. A study by West et al. highlighted a significant association between mROS production and antibacterial response in innate immune cells. This work shows that mROS generation is directly involved in macrophage antimicrobial killing and is mediated by the TLR-TRAF6 dependent signaling [9,31]. 

In addition to mtDNA, neutrophiles can be activated by mitochondrial derived *N*-formyl peptides (mtFP). These peptides resemble bacterial derived *N*-formyl peptides, which bind to G-protein coupled formyl peptide receptors (FPR) and lead to enhanced calcium flux, chemotaxis, oxidative burst, and cytokine secretion [23,32]. 

The antiviral response in innate immune cells is characterized by rapid activation of type I IFN pathways. In this regard, since the discovery of mitochondrial antiviral signaling protein (MAVS) in 2005, the importance of mitochondrial related antiviral signaling has been emphasized in the last few decades [9,33,34,35]. MAVS is located on the outer mitochondrial membrane. Viral double-stranded RNA is sensed in the cytosol by the RLR family like retinoic acid-inducible gene I (RIG-I) and melanoma differentiation-associated gene 5 (MDA5). The detection of viral pathogens activates the caspase-recruitment domain (CARD) of these proteins, which, in turn, interacts with MAVS [9,36]. 

The exact mechanisms through which cytosolic RLRs are recruited to MAVS are still not fully understood and await investigations in vivo [9]. *In vitro*, following activation, MAVS undergoes a conformational change, which is characterized by its aggregation and recruitment of small signaling proteins, such as TNF receptor-associated factor (TRAFs), causing an activation of downstream NF-kB signaling [33]. Initiation of type I IFN pathways through IRF3 and IRF7 signaling is induced by interaction with STING and TBK1 [9,37].

Interestingly, mitochondrial dynamics contribute to effective MAVS signaling. A recent study by Yasukawa et al. demonstrated that MAVS interacts with MFN-2, the protein responsible for mitochondrial fusion. Additionally, inhibition of mitochondrial fusion by knockdown of MFN-2 resulted in decreased NF-κB and IRF3 activation during viral infection [38]. Conversely, depletion of FIS1, the protein responsible for mitochondrial fission, facilitated elongated mitochondrial network, thus leading to improved ER-mitochondrial interactions and adequate virus-induced MAVS-STING signaling [39]. These studies have proven that mitochondrial dynamics and MAVS signaling are important to mount an efficient antiviral response [9,38,39]. 

As noted above, compelling evidence has shown that mitochondria-derived metabolites play a key role in modulating cell phenotype and function of innate immune cells in response to stimuli [1,40]. Macrophages exposed to Lipopolysaccharides (LPS) change to a pro-inflammatory phenotype in vitro, producing high amounts of IL-1β and generating high levels of mROS. These changes are partially mediated by activation of succinate dehydrogenase (SDH), located in the inner mitochondrial membrane, which acts as a Krebs cycle enzyme responsible for succinate oxidization and simultaneously works as a complex II of ETC [1,41]. Succinate metabolism leads to mROS generation and increases HIF-1α-dependent gene expression of IL-1β [41]. Conversely, inhibition of SDH by dimethylmalonate (DMM) reduces LPS-induced mROS production and IL-1β expression. In vivo administration of DMM in mice treated with LPS results in decreased expression of IL-1β and increased generation of IL-10 [1,41]. Additionally, manipulation of SDH in mice infected with *Escherichia coli* or *Salmonella enterica* show a pivotal role of antibacterial host defense in innate immune cells [42]. 

Overall, these studies highlight the ability of mitochondria to adapt and to react during viral and bacterial infections and during sterile inflammation, thus impacting innate immune responses (Table 1) [1,9].

## 4. Interplay between Mitochondria and Adaptive Immune Cells 

In the last few years, several studies have shown that mitochondrial-dependent metabolic state is characteristic for each T cell subtype. Recent findings led to the discovery that pharmacological and genetic manipulation of energy metabolism during an immune response has an impact in mediating cell phenotype and function of T cells [3,40]. 

Naïve T cells are quiescent lymphocytes that utilize nutrients to generate ATP by OXPHOS. Once activated by TCR, they change into an anabolic state associated with high glucose consumption to produce macromolecules required for cell proliferation [3,40]. Both mitochondrial OXPHOS and glycolysis are crucial metabolic pathways to promote T cell proliferation [43,44]. Indeed, mitochondrial function is highly relevant for T cell activation, since an optimal activity of Nuclear factor of activated T-cells (NFAT), NF-κB and proximal T-cell receptor (TCR) signaling is affected by mROS generation [43,45]. Apart from glucose, recent studies have identified glutamine to significantly contribute to mitochondrial metabolism through the transcription factor myc during T cell activation [45,46]. 

Once activated, CD4 T cells can differentiate into regulatory T cells (Treg) or into pro-inflammatory T helper type-17 cells (Th17). When compared to Tregs, Th17 cells are characterized by decreased OXPHOS and increased glycolytic flux [3,47]. Interestingly, the decisive metabolic pathway, shaping the distinctive cell phenotype of CD4 T cells, is linked to the mitochondrial fatty-acid oxidation metabolism [3,47]. Pharmacological and genetic blockade of the enzyme acetyl-CoA carboxylase 1 (ACC1), responsible for the first catalyzation step of de novo fatty-acid synthesis, limits Th17 cell differentiation, and boosts Treg development in vitro and in vivo [48]. In addition to mitochondrial metabolism, manipulation of glucose metabolism alters Treg and Th17 cell ratio, in favor to Tregs, which attenuate experimental autoimmune encephalomyelitis in mice [49]. A study by Shi et al. indicated that the elevated glycolytic flux seen in Th17 cells is mediated by HIF-1α [49]. Furthermore, the HIF-1α target Pyruvate Dehydrogenase Kinase 1 (PDHK1) is highly expressed in Th17 cells, but not in Tregs. Data obtained in vitro and in vivo showed that inhibition of PDHK1 results in decreased Th17 cell number and increased Treg differentiation. These changes were associated with attenuation of Th17 cell induced colitis and EAE central nervous system pathologies [3,50]. 

In addition to CD4 T cells, fatty acid oxidation plays a crucial role in the regulation of metabolism in CD8 T lymphocytes [3,51]. While CD8 T effector cells undergo cell death after infection, some CD8 T cells differentiate into T memory (Tm) cells and stay in a quiescent state, until they get rapidly activated during re-infection. Studies conducted by Pierce and colleagues showed that TRAF6 regulates fatty acid oxidation metabolism to promote CD8 Tm cell differentiation following infection [51]. These CD8 Tm cells exhibit high levels of ATP generation, which is fueled by fatty acid oxidation, a process that generates 3 times more ATP than glucose metabolism [3]. Surprisingly, Tm cells do not increase the fatty acid uptake from the extracellular space, but rather utilize the free fatty acid from the internal cellular storage from lysosomes and promote de novo lipogenesis [52]. 

As discussed, several studies have demonstrated that T cell-fate determination is linked to metabolic programming occurring in the mitochondria (Table 2). However, when compared to the growing evidence highlighting the key role of mitochondrial components in activating signaling pathways in innate immune cells, more extensive investigations should be carried out in the context of adaptive immunity. 

## 5. Mitochondrial Signaling in Trauma and in Critical Diseases 

Trauma is one of the leading causes of death worldwide. Critically ill patients surviving the initial traumatic insult are still at high risk to develop pneumonia and multiple organ failure [53,54].

The activation of the immune system immediately after trauma represents the first line of defense to heal the damaged tissue [53]. These rapid immune responses are mediated by danger molecules released during trauma-induced tissue injury, inducing a pro-inflammatory response, and activating pathways associated with repair mechanism. However, any perturbation of these immune responses including nosocomial infections or immunocompromising comorbidities result in an imbalanced host response [53,55]. Recent investigations on the transcriptome of circulating leukocyte suggest that pro-inflammatory and immunosuppressive responses can occur simultaneously in human trauma patients [56,57]. 

As previously described, mitochondria and associated/derived components can act as DAMPs and modulate the host response affecting sterile inflammation and immunosuppression simultaneously. The first reports defining mtDNA as DAMPs derived from the evidence that trauma patients exhibit high levels of circulating mtDNA, when compared to healthy controls [23,55]. During a traumatic insult disruption of cell membrane results in leakage of mitochondrial components such as mtDNA, mitochondrial derived *N-* formyl peptides or ATP into the extracellular space [55]. These circulating mtDNA and mtFP are sensed by TLR9 and FPR1 expressed on neutrophils [23,55]. Activation of these pathways results in activation, pro-inflammatory response, and chemotaxis of these innate immune subpopulations to the site of the injury [55,58]. Consistent with human studies, in vivo models mimicking the traumatic insult and shock, demonstrated elevated levels of circulating mtDNA, reaching a peak at day 1 and declining afterwards. Increased levels were still noted 7 days after the insult [58,59].

Concordantly, administration of mtDNA to rats induced systemic inflammation and lung injury. Particularly, lung tissues derived from rats injected with mtDNA were characterized by high infiltration of neutrophils [23]. Further, Itagaki et al. have shown that *N*- formyl peptides released from mitochondria during trauma, decrease neutrophil recruitment into the lung while increasing the susceptibility to infection in vivo [60,61]. Suppression of neutrophil chemotaxis was also noted previously in human samples, when looking at the decreased CXC chemokine receptor-2(CXCR2) expression and associated increased incidence of pneumonia after trauma [62]. However, the role of mtDAMPs, including mtDNA and mtFP, was not extensively characterized in these earlier investigations. 

Subsequent investigations established that although mitochondrial components can activate neutrophils, these also suppress migration of innate immune cells to the lung and impede defense against invading pathogens [61]. This was recently demonstrated in a study by Kwon et al. who demonstrated that increased levels of plasma nicotinamide adenine dinucleotide dehydrogenase subunit-6 (the most potent human mtFP) correlated with the development of secondary infection in patients with septic shock. This work indicated that inhibition of neutrophil chemotaxis by mtFP might result in increased susceptibility to secondary infection [63]. These studies highlight the complexity of mitochondrial mediated innate immune responses ranging from inflammatory to immunosuppressive effects in human body.

Our own recent work has provided evidence that, in an in vivo model, whole intact circulating mitochondria, administered prior to bacteria instillation, induce severe lung injury and impair antibacterial host response in mice. Moreover, extracellular intact mitochondria are functionally active in generating ATP [64]. ATP is a well-known danger molecule, which can elicit pro-inflammatory responses by activation of P2X7 and inflammasome signaling [9,27]. To counterbalance this, high levels of extracellular ATP lead to the upregulation of ecto-enzymes CD39 and CD73 with rapid conversion into immunosuppressive adenosine [65]. In our in vivo study, we also showed that mice administered with extracellular intact mitochondria exhibited high proportions of exhausted CD8 T cells, i.e., expressing high levels of CD39, Programmed cell death protein-1 (PD-1) and T-cell immunoglobulin and mucin-domain containing-3 (Tim-3). Notably, human data indicated an association between trauma patients developing systemic inflammation and exhausted CD8 T cell phenotype [64] (Figure 1).

A recently published study by Andrijevic et al. proposed a novel approach based on the combination of extracorporeal perfusion system and cytoprotective perfusate, to restore cellular and molecular function after prolonged warmed ischemia in a porcine model. Hypoxic ischemia is believed to ultimately lead to death. Interestingly, when compared to conventional extracorporeal membrane oxygenation, animals treated with the novel intervention showed reduced tissue injury in the brain, liver, heart and kidney, as assessed by histology. Further, transcriptomic analysis identified enrichment of gene sets positively associated with ATP metabolism and DNA repair when the novel approach was implemented [66,67].

Additional studies have shown that mitochondrial dysfunction can be altered upon transplantation. “Extracellular Vesicle-Mediated Autologous Mitochondrial Transplantation” restored hypoxia-induced alterations of cardiomyocytes and improved post myocardial infarct function in mice [68]. In addition, a study by Fu et al. discovered that mitochondria, derived from young healthy mice, modulate cancer metabolism, and decrease tumor growth. These authors noted that this feature was related, at least in part, to the ability of mitochondria to act as energy suppliers [69]. In a further experimental study, Zhang et al. reported that exogenous mitochondrial transfer could promote bacterial clearance after cecal ligation and puncture, thus inhibiting systemic inflammation and organ injury [70]. 

## 6. Conclusions

This review covers those studies that have shed light into the complex, pleiotropic effects of mitochondria and also discussed the linked ability of these organelles to impact innate and adaptive cell immunity in health and disease. Further investigations are needed to analyze the specific signaling pathways governed by mitochondria in a cell-, tissue- and disease- specific manner. Future work aimed at modulating mitochondria and their composition will hopefully lead to promising therapeutic targets to treat critical disease. 

## Figures and Tables

**Figure 1 cells-11-04113-f001:**
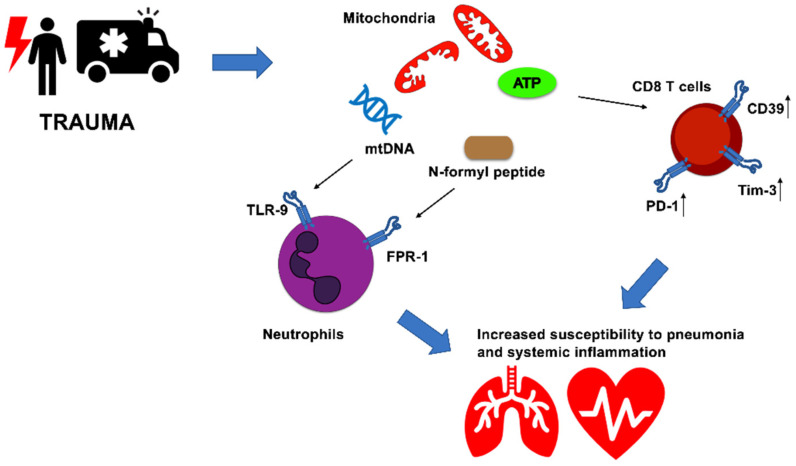
Mitochondria modulate innate and adaptive immune responses and impact clinical course of critically ill patients. Trauma-induced injury elicit rapid immune responses mediated by released mitochondria after cellular damage. Circulating mtDNA and mitochondrial derived *N*-formyl peptides are sensed by TLR9 and FPR1. ATP-producing mitochondria boost CD39, PD-1 and Tim-3 levels in CD8 T cells, favoring immune exhaustion. These immunological changes might increase the susceptibility to secondary infection and systemic inflammation in critically ill patients after traumatic insult.

**Table 1 cells-11-04113-t001:** Interplay between mitochondria and innate immunity.

Signaling Pathways	Components	Response
**Toll-like receptors (TLRs)**		
TLR9	Extracellular mtDNA	Pro-inflammatory response (IL-6, TNF) by NF-kB signaling [23]
TLR1,2,4	mROS	Antibacterial killing by TRAF6 signaling in macrophages [31]
**NOD-like receptors (NLRs)**		
NLRP3	mtDNA	Caspase 1 activation and IL-1β/IL18 secretion by direct interaction with NLRP3 inflammasome [14,24]
	mROS	Caspase 1 activation and IL-1β/IL18 production; mechanism linking mROS and NLRP3 activation unclear [9,28]
	Extracellular mitochondria derived ATP	Caspase 1 activation and IL-1β/IL18 secretion by P2X7 signaling [27]
**Retinoic acid-inducible gene-I (RIG-I)-like receptors**		
Mitochondrial antiviral signaling protein (MAVS)	Viral dsRNA	Activation of MAVS signaling by RIG-I/MDA5 interaction inducing a proinflammatory and robust type I IFN response by TRAF-NF-kB and IRF3/7 signaling in innate immune cells [33,36]
**cGAS-STING signaling**		
	Intracellular mtDNA	Activation of cGAS-STING signaling contributes to type I IFN response in innate immune cells [24,25]
**Formyl peptide receptors**		
	*N*-formyl peptides	Activation of G-protein coupled FPR leads to chemotaxis, oxidative burst, and cytokine secretion in neutrophils [23,32]

**Table 2 cells-11-04113-t002:** Interplay between mitochondria and adaptive immunity.

Adaptive Immune Cells	OXPHOS	Glycolytic Flux	Other Metabolic Pathways
**T cells**			
Naïve T cells	Increased	Increased	Glutamine flux [3,44]
**CD4 T cells**			
Treg	Increased	Decreased	Mitochondrial fatty-acid oxidation metabolism [3,47]
Th17	Decreased	Increased [3,49]	
**CD8 T cells**			
CD8 T effector cells	Increased	Increased [3,40]	
CD8 T memory cells	Increased	Decreased	Mitochondrial fatty-acid oxidation metabolism [51]

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
