# Peer review of "Mitochondria Drive Immune Responses in Critical Disease"

_cells, 2022, doi:10.3390/cells11244113_

Round 1
Reviewer 1 Report
This is a beautifully written review paper on the role of mitochondria in regulating immune responses during critical illness. The authors provide an outstanding overview of state of the art, and the implications on a wide range of critical diseases. I very much enjoyed reading this paper. I only have a few very minor suggestions.
Minor suggestions:
1.) The authors may include a discussion on how hypoxia and specifically HIF1A may affect mitochondrial functions during critical illness (see for example PMID: 28798196 or PMID: 28759485).
2.) The authors may consider expanding on the role of adenosine signaling and its relationship to mitochondrial function, as a downstream product of ATP metabolism (e.g. PMID: 23338058).
3.) The authors may consider including the mentioning of novel and innovative resuscitation approaches, and how those could affect mitochondrial functions in critically ill patients requiring advanced resuscitation approaches (e.g. PMID: 36449428).
Author Response
This is a beautifully written review paper on the role of mitochondria in regulating immune responses during critical illness. The authors provide an outstanding overview of state of the art, and the implications on a wide range of critical diseases. I very much enjoyed reading this paper. I only have a few very minor suggestions.
We thank the Reviewer for the positive feedback.
Minor suggestions:
- The authors may include a discussion on how hypoxia and specifically HIF1A may affect mitochondrial functions during critical illness (see for example PMID: 28798196 or PMID: 28759485).
We have commented on the important links between injury, mitochondrial damage, hypoxia and purinergic signaling on page 3. The suggested references have been included.
- The authors may consider expanding on the role of adenosine signaling and its relationship to mitochondrial function, as a downstream product of ATP metabolism (e.g. PMID: 23338058).
We have added this information on pages 2 and 3 and included the suggested reference. Many thanks for suggesting this important point.
- The authors may consider including the mentioning of novel and innovative resuscitation approaches, and how those could affect mitochondrial functions in critically ill patients requiring advanced resuscitation approaches (e.g. PMID: 36449428).
We have included a comment on this important article on page 8.
Reviewer 2 Report
This is an excellent, and clearly written review. However, it is missing information on the role of the actin cytoskeleton in mitochondrial fusion and fission and the role of mitochondrial fusion and fission in shaping the immune response. There are only a few sentences on this subject, and I feel that a separate paragraph needs to be added.
Author Response
This is an excellent, and clearly written review. However, it is missing information on the role of the actin cytoskeleton in mitochondrial fusion and fission and the role of mitochondrial fusion and fission in shaping the immune response. There are only a few sentences on this subject, and I feel that a separate paragraph needs to be added.
We thank the Reviewer for this suggestion.
We have included this additional information on the role of actin in mitochondria fusion and fission – dynamics - on page 2.
Reviewer 3 Report
The present manuscript is well-organized and comprehensively described article about the interplay between mitochondrial and immune responses. The content is well written, with clear information for readers.
The authors should confirm the small part of the questions listed below.
- Please use a consistent term in the sentence, such as RIG-I- or RIG-I, N-formy peptides, formy peptides, or mtFP
- Please add the full name of the listed abbreviations: IL-6, TNF, IRF3, IFN, NLPR3, P2X7, LPS, NFAT, TCR, FPR1, PD-1, Tim-3, and CXCR2
- Please describe the definition of mtDAMPs and mtFP
- It will be more intact to add the conclusion paragraph at the end of the manuscript.
- Please check the citation reference. There is no reference 62.
Author Response
Reviewer 3:
The present manuscript is well-organized and comprehensively described article about the interplay between mitochondrial and immune responses. The content is well written, with clear information for readers.
We thank the Reviewer for the positive feedback.
The authors should confirm the small part of the questions listed below.
1. Please use a consistent term in the sentence, such as RIG-I- or RIG-I, N-formy peptides, formy peptides, or mtFP
We have changed the abbreviations accordingly.
2. Please add the full name of the listed abbreviations: IL-6, TNF, IRF3, IFN, NLPR3, P2X7, LPS, NFAT, TCR, FPR1, PD-1, Tim-3, and CXCR2
We have added the full name of the listed abbreviations.
3. Please describe the definition of mtDAMPs and mtFP
This definition has been clarified on page 7.
4. It will be more intact to add the conclusion paragraph at the end of the manuscript.
We have revised the conclusion paragraph.
5. Please check the citation reference. There is no reference 62.
Citation references have been checked and updated.